# Requirements for Model-Based Development Process Design and Compliance of Standardized Models

**Avi Shaked \*** and **Yoram Reich**

Systems Engineering Research Initiative, Faculty of Engineering, Tel Aviv University, Tel Aviv 6997801, Israel; yoram@eng.tau.ac.il

**\*** Correspondence: avishakedse@gmail.com

**Abstract:** The planning of system development efforts is crucial to the successful realization of projects. However, development planning typically lacks systematic, engineering discipline, and consequently risks project and business success. Model-based process design is a potential information systems approach to addressing the increasing complexity of such planning. We characterize the ontology of development process design, based on real-life observations and scientific publications. We then synthesize the required ontology with the desirable characteristics of models, and derive key requirements for model-based development process design. Next, these requirements are used to evaluate the adequacy of three prominent, standardized model-based process design approaches—BPMN, OPM and SPEM. The findings reveal that the surveyed models are a partial fit, and do not promote sound process design. Finally, by generalizing the categorical evaluation results, possible root causes for the identified inadequacies are proposed. A new model design, which should rely on the formulated requirements set, is called for, in pursuit of a wider adoption of model-based design paradigms and better information systems realization to support the development of complex systems.

**Keywords:** model-based design; development process design; conceptual modeling requirements; domain ontology

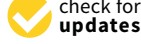



## 1. Introduction

The planning of development efforts is crucial to the successful realization of projects [1,2]. The development of modern, manmade systems becomes increasingly complex, requiring a multitude of activities to be performed as a distributed and coordinated process [3–5]. Furthermore, development activities should relate to multiple perspectives, including project goals, organizational directives and pertinent regulations [6–10]. The identification and coordination of the development activities are best addressed via development planning.

Development planning typically results in recorded plans. Such plans reflect a design of the development process, and are expected to be maintained and managed to reflect performance as the development effort progresses. In practice, development planning primarily focuses on project delivery criteria, whereas it should reflect various aspects of the development effort [1,2]. These findings indicate a profound gap in addressing of multiple perspectives in development plans. It was proposed that adopting systems-related criteria and systems engineering techniques would help address this gap [1,9].

From a systems engineering (SE) perspective, the development process—which designates the engineering perspective of the development effort/project—is the glue that holds together the activities within system development. Theoretically, a development process design (DPD) is expected to ensure the correct application of appropriate development approaches, activities, data, and tools to realize systems and products successfully and

in a controllable manner[1]. DPD, therefore, involves the upfront, timely identification of engineering activities, as well as of the required coordination between them.

An SE approach to addressing the increasing complexity of systems and of their development environment is model-based systems engineering (MBSE) [12]. Computer-aided engineering was shown to contribute to information systems (IS) development [13], and MBSE is an IS-related approach that takes a step forward in the evolution of computer-aided engineering. MBSE relies on the formalized use of computerized databases and models to facilitate engineering design. It can be viewed as a distributed cognition mechanism designated for IS implementations, allowing the design and analysis of large amounts of data, typically using visual representations and a coherent, unambiguous information model [12,14]. Accordingly, MBSE is regarded as an approach that enhances communication between relevant stakeholders, establishes a true shared understanding of the relevant domain, and improves the capture and reuse of knowledge and information. Furthermore, MBSE models are in fact information models that could be represented and used in IS [15]. These models embody relevant knowledge in their structural backbone and procedural definitions; by doing so, they provide rigorous mechanisms for IS and their users to specify, store and validate information. Using standard information models as a technology is critical to the development of IS [16]. It facilitates the sharing of the represented data, via complementary technologies, e.g., by introducing further semantic and syntactic structuring to data represented in the mediating markup language XML. This, consequently, promotes interoperability between different IS, e.g., in cases where a variety of IS are used for different perspectives (alternatively, different views) or between IS implemented by different, even rival, vendors [17]. Additionally, when used in qualitative IS research, information models may support the challenging task of framing questions for data collection, as well as the purposeful analysis of data [18].

Unlike the model-based design of technical systems, which has received significant attention in both theory and practice, the model-based DPD has received little methodical consideration in research, and it is not widely practiced [19] (with professional MBSE tools and IS typically focusing on system design as opposed to process design).

In this paper, we suggest how DPD should be addressed using an MBSE approach. First, we describe three prominent, standardized models ((1) BPMN (business process model and notation), (2) OPM (object process methodology), and (3) SPEM (software process engineering meta-model)) as the state-of-the-art, while mentioning the previously established drawbacks of these models (Section 2). In Section 3, we describe our research method, which focuses on the derivation of requirements for model-based DPD and the evaluation of the competence of models with respect to these requirements. Next, in Section 4, we implement the suggested approach: we derive the requirements for a model-based DPD approach, based on generalizations made by observing development process-related artifacts and the existing literature, as well as by identifying the related, desirable characteristics of process models. Then, in Section 5, we use the established model-based DPD requirements to examine the adequacy of the aforementioned standardized models for model-based DPD. Finally, we discuss our key findings and suggest how they can be addressed in order to provide a model-based DPD solution.

---

[1] The explanation here is a slight adaptation of the description offered by the Modelling and Management of Engineering Processes Special Interest Group of the Design Society, as reported in [11].

## 2. Background

In this section, we provide a brief overview of three process models—BPMN, OPM and SPEM—that are representative of the state-of-the-art process modeling approaches[2]. The criteria for selecting these process models were as follows:

1. The process model demonstrates an MBSE approach by its nature (i.e., includes visual representations of entities from a common data repository, designated for implementation as an IS), and thereby is representative of state-of-the-art approaches to addressing complexity;

2. The process model is documented in a formal specification, which allows us to analyze the model definition objectively as well as attesting to its quality (such specifications are typically released after being thoroughly reviewed by experts, and embody significant domain expertise).

The first criterion filters out traditional approaches that were not adapted to MBSE, such as the once-popular IDEF0 [20]. The second filters out hypothetical models that have yet to be established as prominent IS models. These criteria are similar to the inclusion criteria presented in a recent survey of modeling languages [21]. While our selection does not include all existing approaches, we cover a diverse, representative range of DPD-relevant process models.

BPMN is a notation that can be used for modeling process models. BPMN aims to support the creation of process designs that can be readily understandable to all involved stakeholders [22]. Process modeling is a major focus of business process management (BPM) [23]. Business processes are widely diverse, and while system/product development—a major interest in industrial business—is of an inventive, one-time nature [24,25], BPM is primarily concerned with the core, recurring organizational processes [26]. Still, BPM models may account for the more routine elements of DPD, as well as demonstrating development-related competence and compliance with regulatory standards [27]. BPMN is potentially a good candidate for DPD process modeling for two main reasons. First and foremost, BPMN takes a flow-chart approach to processes, which is also commonly used for describing DPD processes. As such, its use for DPD can inherently cover a comparison with less formal, flow-chart approaches (which are expected to be inferior compared to BPMN use, due to their lack of discipline), as well as the representation of other comparable, flow-chart based approaches (such as activity diagram, which shares the UML foundation of BPMN). In fact, the BPMN standard explicitly mentions a multitude of notations and methodologies that were reviewed in order to consolidate the best ideas into a single standard notation (including UML, IDEF, event-process chains, and others). Second, BPMN clearly identifies that people are "comfortable with visualizing business processes in a flow-chart format," while identifying that there is a gap in the required formality of such process models.

Recker et al. [28] established that BPMN is of high complexity, even when contrasted with the general-purpose language UML [29]. The authors also called for reducing the complexity of modeling methods, and suggested that future research should assist in identifying the gap between theoretical complexity (the entire body of options that exist in the modeling solution) and practical complexity (how a method can actually be used). In their analysis of BPMN as a process model, Dijkman et al. [30] reported that BPMN models are prone to semantic errors, and attributed this to the BPMN specification's ambiguous definitions and eclecticism.

OPM is a conceptual modeling language and methodology for modeling systems [31]. OPM can be used for constructing process models. Sharon et al., for example, proposed a model-based project-planning approach using OPM [32]; and Sitton and Reich used

---

[2]  There are MBSE languages that are designated for other purposes, and are therefore not considered here. For example, SysML (systems modeling language) was devised for the engineering of a technical system, and not for engineering processes. We focus on languages that are specifically designed to address the design of processes (such as BPMN and SPEM—introduced shortly; developed by the same standards consortium responsible for SysML). Unlike SysML, OPM is a general conceptual modeling approach, designed with a significant focus on processes (P stands for process).

OPM as a model-based tool to capture an enterprise-processes framework [33]. However, only theoretical, limited or straightforward implementations of conceptual process models were published, and there are no well-documented guidelines or documentations describing the use of OPM for practical DPD. Li et al., for example, only provided a straightforward model of the V-Model [34]. Similarly, Sharon et al. modeled a simplified, imaginary development project with OPM, explicitly mentioning the lack of states in the model as a simplification [35]. We also add that they did not use OPM for planning (e.g., designing the development process); but, in fact, used it for modeling an existing development process description.

SPEM is a meta-model, which includes a rich language as well as some concepts (relating to methodology), and was designed to "describe a concrete software development process or a family of related software development processes" [36]. In its current version 2.0³, the specification mentions the ambiguous and hard to understand semantics of its previous versions as a reason for their poor adoption. Still, even in its updated form, SPEM remains hard to use, highly complex and hard to understand and implement [38–40].

The aforementioned critique of existing models may provide an explanation for the poor adoption of model-based techniques for DPD. Nevertheless, there is no significant systematic analysis of these models that can provide further insights and suggest how their major drawbacks can be addressed by designing a DPD competent model, which can promote model-based process design (as called for by the likes of Ruiz Rube et al. [39] and Kuhrmann et al. [40]).

## 3. Methods

In our research we derive a set of high-level requirements for a model-based DPD approach (alternately, "DPD model"). The derivation of the requirements set relied on a disciplined SE approach, as follows. We described DPD model characteristics that relate to desirable functionality, by formulating solution-independent requirements. Both DPD ontology and the related modeling concepts were carefully recognized by means of observation and abstraction, as well as by researching published literature. Then, these were synthesized to establish our DPD model requirements set. While the requirements set was deliberately limited (i.e., it is not an exhaustive set) in order to achieve a manageable scope, it is representative of major DPD modeling challenges. Furthermore, following good requirements engineering practices, the requirements were phrased in an abstract way and with respect to the problem domain, in order to avoid concrete limitations on the design space of possible solutions (i.e., the designs of the evaluated models). For example, one of our requirements (as described in the following section) relates to an ability to portray parallel activities, yet it avoids specifying any specific mechanism to do so. This requirement can be satisfied by various designs, such as grouping of parallel activities by additional elements, placement of parallel activities within a specific slot in a notional or concrete time axis, implicit ordering of the activities or their contextual use within parallel workflows.

Then, we evaluated the DPD competence of the models that were previously introduced (Section 2). With our implementation-neutral requirements definitions already providing an infrastructure for proper evaluation of the models, we specified a scoring system, which is described in Table 1. This scoring system allowed us to concisely rate and quantify the compliance of each model with each requirement. The ordinal scoring system reflects the ability to satisfy the requirement using the model, as well as the effort required in order to do so (e.g., does the designer need to invent a compound modeling pattern? Does the model provide well-defined patterns to satisfy the requirement?) and the rigor of a possible application (e.g., does the designer need to select from multiple available modeling patterns? Is there a single pattern that can be applied rigorously by a single

---

³ This current version reinterprets SPEM as "Software & Systems Process Engineering Meta-Model" [37], without explaining the addition of "Systems." Systems, in general, may have characteristics that are different from those of software systems (see, for example, [12], and therefore such a generalization is not to be made lightheadedly.

designer and by multiple designers, to result in comparable models?). We analyzed each of the model specifications with respect to the entire DPD model requirements set. This was performed in order to establish if, how and to what extent a specific model complies with the DPD requirements. This was done based on in-depth reading of the official specification of each of the evaluated models, as well as on an analysis of published research reports. Whenever possible, we incorporated our personal, past experience of using the evaluated model into our assessment.

**Table 1.** Scoring system for evaluating Development Process Design (DPD) model competence.

| Score | Scoring Criteria/Rationale |
|:-----:|:--------------------------:|
| 0 | The model does not comply with the requirement. |
| 1 | The model includes basic support for complying with the requirement, yet significant elaboration and/or alteration is required. Specifically, this score is used whenever relevant modeling constructs can be composed from constructs specified by the model, yet this is prone to user preference and/or requires high expertise. This score may also reflect the lack of a modeling method to support the modeling with respect to the requirement, typically resulting in a cognitive burden on the user, and consequently in idiosyncratic models. |
| 2 | The model implicitly addresses the concept represented by the requirement, yet, it does not specify and/or facilitate rigorous use to support the concept. Specifically, this score is used whenever a model permits multiple semantically distinguished representations of the concept, and/or requires the modeler to select from multiple modeling patterns in order to model a domain-related pattern. |
| 3 | The model explicitly addresses the concept represented by the requirement and/or fully complies with its content. The modeling patterns map to the domain-related patterns with one-to-one correspondence (i.e., there is a bijective function between modeling patterns and domain-related patterns). |

## 4. Requirements Set

In this section, we first present our findings with respect to DPD ontology, and then formulate the requirements for a DPD approach based on the observed ontology. Next, we discuss the related ontology of process models and express it in the form of additional requirements. Finally, we synthesize the DPD approach requirements with the process model requirements, and derive a requirements set for a DPD process model.

### 4.1. Ontology of Development Process Design

By critically reviewing a multitude of system development plans, and based on participation in diverse development projects and environments in both industry and academia, we observed the ontology of development planning from an engineering perspective. Here, we present our observations. In this paper we primarily use the term "ontology" in its original, philosophical sense, as the study of being and the nature of reality, and not in the information systems sense—a manifestation of the philosophical term, which relates to the capturing of domains as formal models (see, for example, [41,42]). Wherever possible, we provide scientific references in support of our observations (throughout this section). Then, we derive requirements for a DPD engineering approach, based on the established ontology.

We identified two prominent conceptual elements in the domain of DPD: artifact and activity. Development processes are typically goal-oriented, and their goal is the creation of artifacts. These observations map well into Braha's network representation of engineering projects (activity as the node, and artifact is the link) [5], and are well aligned with similar observations by Browning [43,44], Munch et al. [27] and Goetschalckx [45]. Specifically, Browning agrees that deliverables (which are a type of artifact) are essential for establishing a system view of the process, and, similarly to our independent observations, he concludes that deliverables are underrepresented in most process models.

Dori et al., made a similar—though more generalized—observation regarding processes in general: a process transforms objects (artifacts) by creating them, consuming them or changing their state [46]. This observation also relates to a further concept we independently observed, despite being less explicit in development plans: artifact-state. When used by a development activity, an artifact is expected to have achieved a certain status, reflecting its readiness to be used in the activity; and similarly, when produced by a development activity, an artifact is expected to reflect a certain state of maturity. This also bears similarity to the concept of deliverables' attributes, proposed by Browning [44][4], and corresponds with another observation by Gericke et al. [11] that using a multitude of attributes is essential to describing a development situation. Correspondingly, the SPEM 1.0 specification identified that "the overall goal of a process is to bring a set of work products to a well-defined state" [36][5]. We observed that process descriptions, in practice, often feature inexplicit artifact descriptions or expectations[6].

The aforementioned concepts provide a systems perspective with respect to the development process. This perspective is aligned with observations made by Simon, who considered process descriptions as a principal form of describing systems, characterizing the world as acted upon, and serving as the means for generating objects that have desired characteristics [47]. While similar, Simon's ontology and the one offered by Dori et al. [46] are not the same: Simon mentioned the "characteristics" of the object, whereas Dori et al. discussed the "state" of the object. Those ontologies can be seen as complementary: a state (Dori) is a time-related concept (a temporal way of being) that can embody a set of stable (though changeable) characteristics (Simon). We refer to this time-related composition of development-related characteristics as the "composite artifact-state." We identified that in DPD, this is indeed the case: artifacts are characterized by achievements made, and these characteristics (also "attributes") collectively reflect the state of the artifacts at each point in time. Understanding the difference between the Dori et al. ontology and Simon's (and, as aforementioned, Browning's [44]), as well as our reconciliation between the two (strictly in DPD context), is key to understanding how ontology may affect modeling.

Some process-related discussions, however, miss the aforementioned systems perspective. Eckert and Clarkson, for example, did not include any reference to input or output objects in their definition of process, regarding it strictly as a set of activities[7] [48]. This compromises the utility of process designs as a bridge between existing and desired states. In fact, even Hallgren and Wilson, who did consider projects as "complex, hierarchical systems", (corresponding with the many ontology-related concepts appearing in this section), only recognized the activities ("tasks") as the "parts" of the system [49].

We observed that plans depicting SE development processes are typically qualitative and procedural. Goetschalckx [45] refers to these as "standard operating procedures." In addition, development plans are typically composed of a non-branching description (i.e., a single depiction of the desired process). This is in agreement with an observation by Gericke and Blessing, who found that development processes are typically described as branch-independent [50], and with Browning, who considered the description of such processes as a "recipe" [44]. Fallback alternatives are typically not captured in development plans, until a decision has been made to incorporate them into the process (based on our experience, this is a good "divide and conquer" strategy that keeps involved personnel focused on the concrete effort while managerial decisions are being made). This nature of development plans differentiates them from other process descriptions that may include further complexity, such as branching (as in computer algorithms, e.g., the well-known

---

[4]　However, Browning's concept relates to the attributes as stationary, not taking into account the evolution of the artifact throughout the development (discussed shortly).

[5]　Interestingly, this important ontological observation is not reflected in SPEM design: immediately after this key observation is mentioned, only "role," "activity" and "work product" are identified as model elements (omitting "state"). Furthermore, in the current version of the specification (2.0), this ontological observation is entirely missing.

[6]　Consequently, this results in miscommunication and poor coordination, which leads to poor cooperation, rework and trouble in development efforts.

[7]　Paradoxically, in the same publication, the authors acknowledge that design processes are unlike workflow-based processes.

"if-then-else" construct) or quantification (as in protocols for experiments in scientific research, such as biological experiment protocol, which typically includes timed activities and measured quantities of materials).

Furthermore, development efforts often include concurrent activities (i.e., activities that occur in parallel) that possibly affect the same artifacts. This is a major challenge that we identified by observing development process designs. Davis and Sitaram [51] and Spillner [52] all noted similar characteristics. We have recorded some related observations from real projects in previous publications [53–55].

We observed that development process design, from a SE viewpoint, is additive by nature. Development activities typically create artifacts and use them to compose additional, higher level systems, as opposed to other process domains that may include the reduction, deconstruction and/or consumption of artifacts (e.g., processes that relate to other stages of the system lifecycle; most notably, system disposal). During a typical development project, subsystems are composed from lower-level components, and these are then integrated to form a larger system. Additionally, the activity description at the SE level typically does not reflect a significant physical transformation of the constituents (e.g., phase transition; manufacturing aspects). Instead, SE level description focuses on the way to compose a system from its constituents, as well as the generation of information (e.g., documentation).

Furthermore, some activities in development processes do not change the actual characteristics of the engineered artifacts (such as testing, verification, review and approval), but are nonetheless fundamental to the project success and to the product maturity (and, in turn, to the organizational/business maturity). Gating activities are an example of such activities. Gating activities serve as a mechanism to inspect and authorize the successful completion of a development project phase, before advancing to the next stage. The case study featured in [54], for example, provides evidence of the existence of gating mechanisms in DPD. Gating activities contribute to the success of development efforts [56]. The designation of the aforementioned activities as system development activities is well aligned with Browning's identification that development activities "gather, create, evaluate, and transform information" [44][8].

We also identified that the design of development processes occurs with respect to a specific scope. The term "scope of work" is well-known in project management, and this has an engineering manifestation. As an example, the designer of the system development process is not the designer of a software item development effort, even if the software is included within the system design, and each of the two designers, therefore, has a different scope of work. While not all process designs explicitly state their scope, we observed that having a well-defined scope is a process design best practice. This is in accordance with Browning's claim that it is essential to specify process boundaries when modeling a process [44]. Furthermore, a scope definition may help in communicating the intent of the process.

The concept of scoping also relates to the established engineering design approach of designing in hierarchies that reflects hierarchical decomposition [45]. This is scientifically sound: hierarchical organization contributes to the communicability, as well to the correctness, of process models [57,58]. Both forward and backward consistency between the different hierarchies of a process design is desirable [45].

Another important aspect of a system development project is that it usually includes a certain level of novelty and creativity [13], and therefore, a certain degree of uncertainty is inherent. This aspect projects onto the development process, which is also characterized by having a degree of uncertainty, and often requires the tailoring of past knowledge and methods to a specific design situation before they can be applied [7,37,44,48].

---

[8] Browning further stresses that not all development related activities should be incorporated into a process plan, and specifically mentions that general infrastructure activities are best left out. Overall, we agree with this approach.

Furthermore, some of the uncertainty can be attributed to the need for delegating authority to involved teams, in order to encourage their commitment to the project and support pertinent decision making. This requires a mechanism that balances the need for process control and the need for accommodating degrees of freedom of process design [9].

Consequently, the development effort is iterative, and its details unfold as it progresses [50,59]. The development course, and particularly advancements in engineering knowledge (which may be the result of investigating a failure to meet a desired goal or to produce an artifact, or just a better understanding acquired throughout the effort), typically require revisions to be made. Unfortunately, we observed that—in practice—development plans tend to be rather static, and that DPD is not being practiced continuously throughout the development effort. This, in turn, can be attributed to a gap in the state-of-the-art identified by Karniel and Reich: process models are assumed to be static [59].

We now summarize the desirable properties of a DPD approach based on the ontology, as observed and established.

O1. DPD shall involve the following entities for process composition: activity, artifact, and composite artifact-state.
O2. DPD shall support a qualitative, non-branching, procedural description of activities.
O3. DPD shall support detailing parallel activities.
O4. DPD shall allow for describing the additive/constructive perspective of development activities.
O5. DPD shall reflect a process scope.
O6. DPD shall encourage a hierarchical (multi-level) approach. Additionally, this further establishes that DPD shall support forward and backward consistency between the different hierarchies of a development plan.
O7. DPD shall be able to accommodate changes as well as uncertainty and purposeful creativity. Additionally, this further establishes that DPD shall incorporate a mechanism to allow control of the process, while offering degrees of freedom for its design.

In the next subsection we derive requirements for process models, before synthesizing them with the aforementioned DPD approach requirements to result in requirements for a model-based DPD approach.

### 4.2. Related Ontology of Process Models

Process models are associated with three main aims: being descriptive of actual occurrence; bring prescriptive, defining desired processes; and being explanatory, providing rationale for the processes [60]. Process models can be used in order to describe and orchestrate the development activities in the context of the development process.

It is important to distinguish a process model from the actual process. A process model is a rough anticipation of what the process will look like, or, alternatively, is a purposeful abstraction of the actual process [48,60]. Rolland addressed the following desirable characteristics of a process model [60]:

M1. Process models shall, ideally, provide a wide range of granularity.

Rolland attributed granularity to the kind of guidance, explanation and trace that can be provided, and mentioned that this granularity support shall allow movement from large grains to fine grains along a continuum;

M2. Process models shall provide a mechanism to accommodate change.

The model shall allow for recording changes and emergent occurrences, during and/or after process execution, to reflect updated process design and experience gained.

Acknowledging that development processes differ one from the other, Eckert and Clarkson [48]) claimed that a plan "can only be drawn from experiences of similar projects", (p. 154), as opposed to reusing the design of a previous development process in its entirety. Accordingly, they suggested that development process models should support the reuse of elements of previous models. Browning also called for such reuse, and stressed its importance for capturing and managing development process-related knowledge, as well as for the absorption of best practices [44]. Method engineering is a discipline that further builds on such concepts, as it is concerned with the design, construction and adaptation of methods for the development of systems [61]. In fact, method engineering can be regarded as a specialized branch of DPD, as it aims to generalize process designs (either in their entirety or by extracting segments of these designs) into reusable method components that can be used in new designs, as well as detailing how the tailoring of components into new designs can be performed. As such, method engineering can benefit from process models that allow one to capture process segments and reuse them as building blocks in new process designs;

M3. Development process models shall enable the reuse of elements of previous models that were used to design similar components or systems.

Eckert and Clarkson referred to the aforementioned abstractive nature of models, and recommended that one considers what to reflect in the model and what to leave out. Correspondingly, Wand and Weber provided criteria for evaluating IS analysis and design methodologies, and—in what became known as the Bunge–Wand–Weber model—they stressed the importance of the ontological clarity of language constructs, i.e., the clear association of a language construct with an ontological construct [41];

M4. Process model shall clearly correspond with related process ontology.

That is, the technical, IS-related model implementation should clearly correspond with the ontology of the real-life domain that we wish to model.

*4.3. Model-Based DPD Requirements*

By synthesizing the DPD approach requirements (of Section 4.1) with the process model requirements (of Section 4.2), we derived the following set of requirements for a DPD process model.[9] Each bulleted text represents the synthesized requirement, with the identification of its originating requirements (from the aforementioned sets) listed in brackets. Each of the synthesized requirements is then represented by one or more concise requirement definition(s), designated with a REQ prefix for use in the subsequent section.

- The DPD model shall include the following entities for process composition (activity, artifact, composite artifact-state [O1,M4]):

REQ1. Support Activity representation;
REQ2. Support Artifact representation;
REQ3. Support Composite Artifact-State representation.

  - DPD model shall support a qualitative, procedural description of activities [O2, M4]:

REQ4. Support a procedural description of activities.

  - DPD model shall support detailing parallel activities [O3, M4]:

REQ5. Detail parallel activities.

  - DPD shall allow describing the additive/constructive perspective of development activities [O4, M4]:

REQ6. Describe the additive perspective of development activities.

---

9    These requirements are high-level functional requirements, and can be elaborated with both lower-level functional requirements and non-functional, performance-related requirements. Since our goal here is to develop and discuss DPD modeling concepts, this level of requirements should be sufficient, and whenever we feel it is appropriate, we provide a discussion with respect to lower-level details.

- DPD model shall reflect process scope [O5, M4]:

REQ7. Reflect process scope.

- DPD model shall encourage hierarchical decomposition of a process, and provide consistency mechanisms [O6, M1]:

REQ8. Support consistent representations of process hierarchies.

- DPD model shall be able to accommodate changes as well as uncertainty [O7, M2]:

REQ9. Accommodate changes and uncertainty.

- DPD model shall allow, and preferably promote, reuse of previous models' elements [M3]:

REQ10. Support reuse of previous models' elements.

## 5. Evaluation of Modeling Approaches Compliance with DPD Model Requirements

This section presents our evaluation of the three state-of-the-art process modeling approaches—which were introduced in Section 2—with respect to the DPD model requirements. Each of the following subsections is dedicated to a specific model, and presents its competence evaluation. Our evaluation begins with the main analysis highlights, and concludes with a table detailing the requirement-specific assessment, which includes a competence score as well as a brief reasoning of our assessment. While we made a significant attempt to simplify the assessment language, some of the reasoning may require the reader to be familiar with the evaluated model, as it is impractical to explore the complex modeling standards here.

### 5.1. OPM Evaluation for DPD

Table 2 summarizes our evaluation of OPM for DPD, based on analysis of the ISO/PAS standard [31], existing literature[10], and personal experience of using OPM. Key points of our assessment follow.

While OPM inherently features desired modeling entities to represent DPD entities (activity, artifact, artifact-state), OPM does not support the ability to represent a composite artifact-state. OPM does relate to "object states", but allows an "object" to be in only one explicit specific state [46], which is insufficient to address the complexity of our domain. Accordingly, there is no inherent mechanism to portray the constructive perspective of development activities.

OPM provides a diagram-based mechanism for hierarchical process representation, but this mechanism does not support establishing forward and backward consistencies between the different hierarchical levels by design. Accordingly, this mechanism does not reflect process scope, as it is allowed to exhibit partial scope in various hierarchies. This also hinders the reuse of models or model fragments. Furthermore, OPM is a general-purpose modeling framework, and accordingly, its use is prone to personal preferences, promoting idiosyncratic models that are not easily comparable and do not allow for transfer of knowledge [32]; we therefore deem the OPM methodology as not promoting reuse[11].

### 5.2. BPMN Evaluation for DPD

Table 3 summarizes our evaluation of BPMN for DPD, based on analysis of the current BPMN 2.0 specification [22] and existing literature[12]. The key points of our assessment follow.

---

[10] Including [21,32–35,46,62–64].

[11] As in the agreement, the OPM standard [31] does not mention "reuse" at all.

[12] Including [21,30,65–67].

**Table 2.** OPM evaluation against DPD model requirements.

| Requirement | Model Support Score | Brief Reasoning |
|---|---|---|
| REQ1 Support Activity representation | 3 | "Process" element |
| REQ2 Support Artifact representation | 3 | "Object" element |
| REQ3 Support Composite Artifact-State representation | 1 | A "state" modeling element exists, but no inherent method to support a composite state of the artifact, as OPM allows an object to be in one state at a specific time. In addition, OPM accepts an alleviative approach that does not necessitate using state designations in process descriptions. |
| REQ4 Support a procedural description of activities | 1 | A sense of flow/orientation can be established, but is not a focus of the model. An implicit arrangement of activities (top to bottom) can be used to depict their order of execution, but this is used primarily for simulation purposes (and, in turn, risks the ability to simulate concurrent activities). |
| REQ5 Detail parallel activities | 3 | Activities can be modeled regardless of execution order. |
| REQ6 Describe the additive perspective of development activities | 0 | No inherent support. |
| REQ7 Reflect process scope | 1 | Process scope may be established manually, but is not self-contained within a representation and/or emerges out of the representation. |
| REQ8 Support consistent representations of process hierarchies | 2 | Multi-level representation of the process hierarchies is available via diagrams "in-zooming" and "unfolding", but no consistency is assured. For example, the "in-zooming" diagram is used to present two hierarchies of the process (the process and its sub-processes), and allows to model relations between sub-processes and external entities without requiring these relations to be specified in the higher, process level hierarchy. Specifically, forward and backward consistency is hindered by support of dissimilar relationships in various levels of hierarchies. |
| REQ9 Accommodate changes and uncertainty | 2 | OPM supports encapsulation for processes. However, its encapsulation does not seem to be rigorous enough, and may be open to interpretations. Furthermore, OPM does not promote consistency between diagrams of different hierarchies: when changing links for an activity in one hierarchy, its lower and/or higher hierarchies are indifferent to the change. |
| REQ10 Support reuse of previous models | 0 | No DPD modeling method exists, and this results in idiosyncratic process models. In turn, this hinders reuse of elements from previous models, especially since they are not available as a self-contained representation. |
| **Total Score** | **16/30** | |

**Table 3.** BPMN evaluation against Development Process Desgin (DPD) model requirements.

| Requirement | Model Support Score | Brief Reasoning |
|---|---|---|
| REQ1 Support Activity representation | 2 | "Activity" element exists. However, there are redundant representations of activities in the form of "Sub-process" (for non-atomic activities) and "Task" (for atomic activities), which are considered extended elements. Additional elements may hinder correct modeling: "Group" <br> Task Types: "Service Task", "Send Task", "Receive Task", "User Task", "Manual Task", "Business Role Task", "Script Task", "Call Activity" <br> Sub-process Types: "Transaction Sub-Process", "Ad-Hoc Sub-Process", "Call Activity" |
| REQ2 Support Artifact representation | 3 | "Data Object" element exists. |
| REQ3 Support Composite Artifact-State representation | 0 | BPMN allows only a single state (using the optional "dataState" attribute) to be attached to a data object element. There is no inherent support for artifact's composite state representation. |
| REQ4 Support a procedural description of activities | 3 | A sense of flow/orientation can be established, using flow elements. |
| REQ5 Detail parallel activities | 3 | Supported. This is explicitly identified as one use case of using "Sub-process" entities. |
| REQ6 Describe the additive perspective of development activities | 0 | No inherent support. Artifacts are considered optional. |
| REQ7 Reflect process scope | 1 | Process scopes may be established, and are hierarchically nested. A scope is defined by the specification as a set of data objects available, events and conversation. However, the graphical notation does not reflect the scope (especially with respect to data objects), and the model seems to relate to the scope as an execution mechanism rather than a design mechanism. "Sub-process" modeling entities are explicitly mentioned as a mechanism to design a contextual scope, but this does not relate to artifacts in the establishing of such scope. Specifically, activities ("sub-process"/"task") may implicitly use artifacts ("data object") of higher hierarchies, hindering understanding of their real scope. |
| REQ8 Support consistent representations of process hierarchies | 2 | Multi-level representation of process hierarchies is available (using "Sub-process" entities), but no consistency issues are addressed. |
| REQ9 Accommodate changes and uncertainty | 1 | "Task" entities may be used to accommodate uncertainty. The models, however, exhibit low tolerance to changes, and specifically do not methodically address the propagation of changes. |
| REQ10 Support reuse of previous models | 1 | A process can refer to another process via the "Call Activity" mechanism, yet this "black box" call to a process does not contribute to a disciplined process design. No DPD modeling method exists, resulting in idiosyncratic models. In turn, this hinders the reuse of segments from previous models. |
| **Total Score** | **16/30** | |

BPMN features the desired modeling entities to represent the DPD entities "Activity" and "Artifact", but does not inherently include representation of "artifact-state." Artifacts—represented by "Data Objects"—are considered optional, and this does not promote a sound DPD practice. Interestingly, an activity has redundant representations. This is but one example of the redundancies that exist within the BPMN specification. While these redundancies were most likely devised in order to present a rich language, they may hinder the use of BPMN, as they complicate usage and facilitate idiosyncratic modeling (and, in turn, discourage the reuse of models).

With BPMN being primarily a modeling notation, it does not provide a method to support DPD using the notation. This is notable in BPMN's complete neutrality to the constructive nature of DPD and to the artifacts as part of a process scope. Similarly, there is no suggested method to promote consistency between the various levels of the process model.

*5.3. SPEM Evaluation for DPD*

Table 4 summarizes our evaluation of SPEM for DPD, based on analysis of the current SPEM 2.0 specification [37], existing literature[13], and personal experience of using SPEM. The key points of our assessment follow.

**Table 4.** SPEM evaluation against Development Process Desgin (DPD) model requirements.

| Requirement | Model Support Score | Brief Reasoning |
| --- | --- | --- |
| REQ1 Support Activity representation | 2 | "Activity" element (a "Work Definition"-derived entity). Potential redundancies: "Milestone"; other "Work Definition"-derived entities: "TaskDefinition", "Step", "Task Use", "ProcessComponent", "ProcessComponentUse" |
| REQ2 Support Artifact representation | 2 | "Work Product Definition" "Work Product Use" element. This element is an activity-specific object, and therefore does not support artifacts outside of activities. |
| REQ3 Support Composite Artifact-State representation | 1 | States can be detailed explicitly (extending UML state machines and activity diagrams to support this). However, there is no inherent mechanism for artifact composite state representation. |
| REQ4 Support a procedural description of activities | 1 | A sense of flow/orientation can be established. The abundance of entities, relations and diagrams, however, significantly hinders such description. The standard acknowledges that the UML-derived representation is not practical, and suggests a structured tabular, textual representation ("Work Breakdown Structure Presentation") to be used instead. The tabular representation does not facilitate process orientation. |
| REQ5 Detail parallel activities | 2 | SPEM supports and encourages the modeling of activities regardless of execution order. However, it seems that process design is somewhat left to enactment itself (which is out of the scope of SPEM methodology) to make such design decisions. |
| REQ6 Describe the additive perspective of development activities | 0 | No real methodology concerning this, but rather a collection of various, alternative mechanisms. UML-based state transitions, various implementations of activity diagram, and attributes of both activity and artifact elements may be used, promoting idiosyncratic modeling. |

---

[13]  Including: [27,40,68–73].

**Table 4.** *Cont.*

| Requirement | Model Support Score | Brief Reasoning |
|---|---|---|
| REQ7 Reflect process scope | 1 | A partial process scope may be established manually (by analysis of models/representations), but is not self-contained within a representation and/or emerges out of a representation. Additionally, leaving out some design decisions to process enactment reduces the ability to recognize scope. ProcessComponent provides a good scoping mechanism for encapsulation. However, it does not facilitate the rigorous definition of artifacts (one example does refer to the use of a UML "state invariant" element to depict such states[14]), and no proper methodological attention is given to such use of scoping (reading the standard, ProcessComponent is given a limited attention). Furthermore, the specification strangely states that this element contains "exactly one Process that is physically encapsulated." |
| REQ8 Support consistent representations of process hierarchies | 2 | Multi-level representation of process hierarchies is limited, and concentrates mainly on a breakdown structure mechanism to define the organization of methods and activities (including tasks and steps, which are not interchangeable). Consistency is not assured. ProcessComponent may provide a good mechanism for this; but its use needs to be detailed to provide rigorous use of this element. |
| REQ9 Accommodate changes and uncertainty | 1 | A separation between method content and its complex reuse mechanism (e.g., various element types) for process design hinders model changes. While "Activity" element was conceived for supporting uncertainty (or self-organizing teams, in the domain of software engineering for which SPEM was developed), the details of reusable methods (specifically "task" and "step") reduce the ability to accommodate uncertainty. "Process Component" provides a good encapsulation mechanism, but is designated by the standard for use in situations of delegation or undecided black-box activity (and not as a hierarchical, SADT-like process design mechanism). |
| REQ10 Support reuse of previous models | 1 | Reuse of method content is encouraged, provided it is fully reused; with mostly the sequence of using method content left as a degree of freedom, for a specific project design. The separation between method content and actual process composition hinders the post-completion reuse of performed activities design as future method content. The details of reusable methods and the redundant definitions may hinder their effective reuse, especially in different hierarchies. |
| **Total Score** | **13/30** | |

SPEM features desired modeling entities to represent all of the DPD entities ("Activity", "Artifact" and "Artifact-State"). However, while SPEM is more development process-oriented than BPMN, it seems to include an increasing amount of redundancy,

---

[14] This is not trivial. The UML standard [29] specifies "State Invariant" as a constraint evaluated at runtime (i.e., with respect to our domain, it is not evaluated upon design/analysis). The UML standard considers "State Invariant" model entity as an "InteractionFragment" entity to be placed on a lifeline (within an activity diagram), and this is also the only mechanism of using this element shown in the UML specification. A SPEM brief example of using such an element shows an unspecified extension of the UML meta-model.

and by reading the specification, one can easily attribute this redundancy to a desire to satisfy various approaches to the practice, without providing a clear, ontology-relevant methodology for DPD.

The redundancy manifests through an abundance of possible modeling elements and representations. The specification itself provides support for this assertion, by recommending a hierarchically-arranged, list-styled "Work Breakdown Structure Presentation", as a practical alternative to its UML-based method content diagram[15]. This, however, does not promote the procedural description of activities, and especially makes it hard to establish a sense of flow.

SPEM differentiates method content definition from process composition (which uses method contents). This is done—supposedly—in an attempt to encourage the reuse of method content in various processes. SPEM also differentiates "activity" from a lower-level "task" and an even lower-level "step". These do not provide the desirable transition between modeling hierarchies ("resolutions") along a continuum. All of these hinder the reuse of elements in DPD scenarios (that typically require tailoring of past processes), and do not provide a mechanism that accommodates changes well[16].

A basic hierarchical representation of a development process exists in the form of a breakdown structure mechanism for activities, which does not promote any consistency. However, a thorough investigation of the SPEM specification reveals a mechanism that is suitable for the multi-level representation of a development process and may promote consistency—process component. Process component was introduced in SPEM 2.0, and has the potential to provide a solid hierarchical representation, as well as supporting other requirements (such as reflecting process scope and accommodating changes and uncertainty). However, the specification falls short in the proper description of the process component and how it can be used as a holistic DPD approach[17].

*5.4. Models Evaluation Summary*

Table 5 summarizes the scores of the three models. The models fall short of providing support for DPD. The major deficiencies are in requirements 3, 6, 7, and 10.

**Table 5.** Summary of models' evaluation against DPD model requirements.

| Requirement# | OPM | BPMN | SPEM |
|:---:|:---:|:---:|:---:|
| 1 | 3 | 2 | 2 |
| 2 | 3 | 3 | 2 |
| 3 | 1 | 0 | 1 |
| 4 | 1 | 3 | 1 |
| 5 | 3 | 3 | 2 |
| 6 | 0 | 0 | 0 |
| 7 | 1 | 1 | 1 |
| 8 | 2 | 2 | 2 |
| 9 | 2 | 1 | 1 |
| 10 | 0 | 1 | 1 |
| **Total Score** | **16/30** | **16/30** | **13/30** |

---

[15] This recommendation does not promote the desired SE approach to DPD, as it does not include artifact-related data. The specification also mentions that this approach is "utilized by popular project planning tools," further attesting to its managerial perspective origin.

[16] SPEM acknowledges the need to perform tailoring for specific project situations, and provides mechanisms that support tailoring for "method content" and "process patterns." These, however, require high user proficiency, and do not accommodate changes.

[17] Process component usage is depicted for some specific situations (mostly for encapsulation of work), and is not encouraged or described sufficiently to be used as a general design paradigm. Furthermore, the description of the associated "invariant" modeling entity is insufficient; especially considering that its use is an extension of the UML standard [29], which remains undefined. Accordingly, we found that a leading MBSE commercial tool that is SPEM 2.0-compliant—Sparx Enterprise Architect—does not support this.

## 6. Discussion

As system development becomes increasingly complex, a model-based approach to handle the design of the development effort is crucial. However, current model-based approaches are lacking with respect to DPD needs.

Here, we established a set of high-level requirements for model-based DPD to reflect the domain-specific modeling needs. This set was synthesized from a set of requirements that we formulated for a DPD approach—based on the pertinent ontology—and a set of process model requirements, which was based on the identification of the desirable characteristics of process models. Our requirements set reflects extensive theoretical concepts as well as practical experience. We encourage future research efforts to use our solution-neutral requirements set as a basis for the design of new model-based approaches, and address the gap in effective model-based DPD.

We used the DPD model requirements to evaluate prominent modeling specifications. Our evaluation of the specifications indicates a significant gap in the usability of current IS approaches for DPD modeling. Specifically, these state-of-the-art modeling approaches fail to directly acknowledge the complexity of development artifacts and the constructive nature of the development process. Failing to address such key ontological elements is problematic. Current approaches also lack in representing the scope of processes, and—possibly consequently—lack in providing mechanisms that promote the reuse of segments from a process model. The identified DPD modeling usability gap may explain the reported poor adoption of model-based DPD approaches. Accordingly, this attests to the utility of the established DPD model requirements.

Furthermore, the scoring system that we devised for the evaluation of the models can be perceived as a synthesis of our requirements set with another set of model requirements. As opposed to a binary system of evaluating the compliance of a design with requirements (satisfied/unsatisfied), our scoring system also allows us to evaluate the quality attributes of implementation more objectively, while remaining design-neutral. While a score of 0 indicates no inherent support for the specific requirement, a score of 1 indicates a basic ability to satisfy the requirement but without a specified modeling method; a score of 2 indicates the existence of redundant representation or approaches that may risk the model utility and/or integrity, in accordance with the expected ontological clarity (Bunge–Wand–Weber model); and a score of 3 represents a design that is ontologically clear and, if applicable, supported by a specified modeling method. For example, BPMN's lack of inherent support for representing artifact-states resulted in a score of 0 for REQ3. OPM scored 1 for the same requirement, since it does support the artifact-state representation to some extent, yet with no specified method to introduce such states to the model in a way which is aligned with the domain. Another example is SPEM's full support of the ability to represent the activity ontological concept, and yet its score for REQ1 is 2, as it allows multiple ways to do so, whereas OPM has only one representation for activity, and therefore its score for REQ1 is assessed as 3.

Our findings suggest several additional possibilities for future research. Model design should satisfy the lifecycle of using the model, and not just its static structure, and should therefore incorporate holistic considerations. Specifically, IS research which aims to provide a sound DPD model should develop such a model in accord with domain-relevant ontology, as well as with the modeling objectives. Our DPD model requirements set can provide a strong basis for such alignment, as it synthesizes the observed DPD characteristics (for example, involved entities) with desirable model characteristics (for example, consistent hierarchical structuring and scoping). Mapping data into model elements contributes to the ability to draw configurations, as well as to analyze data for consistency and for completeness [18]. A well-structured DPD model, which satisfies the derived requirements, may support the rigorous design and analysis of development processes. We intend to develop such a model, while also considering how it can be integrated with additional perspectives, such as a managerial perspective and a contractual perspective that emphasizes the intent and expectations from a development process.

Further research may generalize and build on our analysis and method for assessing the competence of, and improving, existing or new model-based design methodologies for other domains (e.g., system architecture, cybersecurity assessment, or project management) based on specified requirements, for the purpose of introducing standardized information models that can be used effectively in information systems.

**Author Contributions:** Conceptualization, A.S.; methodology, A.S. and Y.R.; analysis, A.S.; writing—original draft preparation, A.S.; writing—review and editing, Y.R.; visualization, A.S. and Y.R.; supervision, Y.R. All authors have read and agreed to the published version of the manuscript.

**Funding:** This research received no external funding.

**Informed Consent Statement:** Not applicable.

**Data Availability Statement:** Data sharing not applicable.

**Conflicts of Interest:** The authors declare no conflict of interest.

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
