# Peer review of "Requirements for Model-Based Development Process Design and Compliance of Standardized Models"

_systems, doi:10.3390/systems9010003_

Round 1

Reviewer 1 Report

Model-based process design has been considered to address the complexity of development planning.
An ontology for the development process design is specified, then characteristics of models are introduced. On the
basis of these two results key requirements were derived and used to evaluate three prominent model-based process design approaches.
The authors conclude that the approaches under consideration are a partial fit and a new model design is called for.

This paper is well structured and clearly presented. The problem this work tries to address is important, and the idea of applying MBSE
approaches seems promising. However, many issues should be addressed:

Line 92:
I know there are many process modeling languages and methodologies out there and the authors considered those that fit their criteria. It
is not clear to me why SysML is not included. It is pretty dominant. Authors should consider it or at least justify its absence.

Line 107:
The authors introduce the MBSE approaches superficially. I was expecting to have the features and the corresponding notations that the
authors consider in their evaluation section so the reader can connect between what the approach has and the arguments of the authors. I know it
is not practical to present all three approaches completely, but the paper does not include enough to help the reader understand its content.

Table 1:
Table 1 introduces the scoring system used for MBSE approaches evaluation.
To me, the second criteria is too simplistic and does not honestly reflect what it is supposed to. There are MBSE approaches that
are defined with minimal yet universal ontology, which means that new concepts are built from its basic building blocks. Such approaches
are very easy to learn (learning cure is steep and short) and the burden will be in constructing new concepts from existing ones... even
here sometimes the composition is pretty straightforward and it is worth a higher score (>1). I think the score should be measured by the
effort spent - number of basic blocks needed to compose a new one or depth of composition. etc. Further, we do not expect to have
an MBSE approach the has a perfect fit, so a threshold should be specified for good compliance. In other words, the total scores
should be translated into qualitative representation.

191:
I'm missing an elaboration on the methodology of this section. You are providing ontology and properties for the DPD approach
but it is not clear how you got that. This will make hard to replicate this work or adopt your methodology in future work.

Line 218:
In OPM, every object can have (exhibit) attributes using the exhibition-characterization relation. An attribute by itself
is considered an object which in turns can have states which are called values. Based on this, the assertion that
OPM does not support object characteristics is not clear. Please elaborate on characteristics of objects according to Simon and how they
are different from those of OPM.

General:
I think an example or real (partial) models throughout the paper would make things clearer. It saves text and explains much better.

401:
In "Model-Based Systems Engineering with OPM and SysML" book there is a section dedicated for the compound states. Further more,
paths and path labels can be used to defines disjoint states. OPM also defines State-Specified Tagged Structural Links that show how
an object (with states) can be a state. I think these OPM mechanisms should be considered in the paper by accepting or declining them
as part of the suggested solution.

405:
OPM supports hierarchical decomposition using in-zooming and unfolding. It supports also out-zooming if there is a need to create
another level upward. Consistency is preserved between the different levels (maybe partially).

407:
It is not clear to me why this mechanism does not reflect scoping. OPM supports scoping by allowing processes and objects to be environmental.

Table2Row4:
According to the OPM timeline principle subprocesses in an inzoomed process are executed from top to bottom providing a clear flow.

T2R6:
OPM outzooming is a kind of building the system from lower levels upward. This is not to say that OPM inherently support this property
but it does that in some way.

T2R8:
To the best of my knowledge, OPCloud, the cloud-based implementation, supports that - changing a link in an inzoomed process
will propagate to other levels. It works in both directions.

T2R9:
Same as T2R8

417:
In BPMN, Data object elements can optionally reference a datastate element which is the state of the data contained in the data object.

T3R10:
BPMN call activity allows to create a reusable process definition, Partial fit?

474:
It is not clear what kind of approach is acceptable? Should it be a perfect fit? What score should it achieve to be acceptable?

Reviewer 2 Report

This paper deals with the issue of modeling system development processes. It uses an ontological approach to identify salient characteristics of such processes and proposes a set of requirements to be fulfilled by any process model to match these characteristics. It finally shows that three of the most referential modeling approaches are far to meet these requirements.

The paper has a number of strengths:

  • The topic is relevant and important because the state-of-practice shows a predominance of hand–crafted approaches far to be satisfactory in terms of quality and efficiency.
  • The non-adoption of any of the existing system development process meta-models & methodologies (including such formalized models to drive the development processes), poses questions that the paper adequately considers.
  • Deriving requirements from an ontological analysis is valid.
  • I find that all identified requirements (REQ1 to 10) are correct and that they point out very important dimensions of development processes, which are too often ignored.
  • For example, the identification of the relationship process-product (REQ1, 2, 3) is fundamental. The predominance of approaches viewing processes as simple sequences of activities without reference to the state of the product they change must be pointed out as an erroneous way-of-working.
  • Similarly, reuse of ‘best practices’ (REQ10) is inherent to efficient development processes and shall be promoted.
  • Recognizing uncertainty (REQ9) is important as it suggests a shift from a procedural approach of development processes whereas they are fundamentally decision-driven processes.
  • Tables 2,3, 4 and 5 are useful to convince stakeholders of development processes that they should reconsider the predominant activity-based view of their processes.
  • The paper is clearly written, easy to follow and understand.

Here are also some suggestions for improvement:

  • Despite the suggested shift from activity centered modeling to activity-product modeling (which is well appreciated), I would suggest to go further and move towards a decision-centered view.
  • Fundamentally the development process engineer is repeatedly making decisions confronted to situations with multiple choices. At that level, the concept of goal/intention is relevant to catch what the engineer would like to achieve. To achieve the goal, there are multiple options with pros and cons. The essence of the development process is not only to perform actions to change the state of the product (artifact) but to take actions as a consequence of a decision made in a multiple-choice situation. In summary, there are three meta-meta modeling views: activity; activity-product; goal- activity-product. Whereas this paper promotes the second, I would suggest extending the requirements set to capture the intentional/decisional view of processes.
  • Following this line, method engineering is a domain in which different dimensions of the development process have been investigated (including intentionality of processes). I suggest to further consider researches of this domain and to include associated references.
  • A minor comment is related to the use of the term ‘ontology’. It seems that what the paper does is more to follow an ontological analysis than to strictly build ontology.

Globally, the paper advances our knowledge on the field of modeling system development processes in a significant way. If possible, mentioned extensions should be considered in the final version of the paper.
